# `scCLIP`: Multi-modal Single-cell Contrastive Learning Integration Pre-training

**Lei Xiong**,* **Tianlong Chen***, **Manolis Kellis**
Massachusetts Institute of Technology
{leixiong, tianlong, manoli}@mit.edu

## Abstract

Recent advances in multi-modal single-cell sequencing technologies enable the simultaneous profiling of chromatin accessibility and transcriptome in individual cells. Integration analysis of multi-modal single-cell data offers a more comprehensive understanding of the regulatory mechanisms linking chromatin status and gene expression, driving cellular processes and diseases. In order to acquire features that align peaks and genes within the same embedding space and facilitate seamless zero-shot transfer to new data, we introduced scCLIP (**s**ingle-**c**ell **C**ontrastive **L**earning **I**ntegration **P**retraining), a generalized multi-modal transformer model with contrastive learning. We show that this model outperforms other competing methods, and beyond this, `scCLIP` learns transferable features across modalities and generalizes to unseen datasets, which pose the great potential to bridge the vast number of unpaired unimodal datasets both existing and new data generated in the future. Specifically, we propose the first large-scale transformer model designed for single-cell ATAC-seq data by patching peaks across the genomes and representing each patch as a token. This innovative approach enables us effectively to address the scalability challenges posed by scATAC-seq, even when dealing with datasets of up to *one million* dimensions. Codes are provided at: https://github.com/jsxlei/scCLIP.

## 1 Introduction

Single-cell sequencing techniques such as single-cell RNA sequencing (scRNA-seq) [1] for transcriptomic and single-cell assay for transposase-accessible chromatin with sequencing (scATAC-seq) [2, 3] for epigenomic have emerged as indispensable tools for dissecting the intricate mechanisms of cellular heterogeneity and functionality in diseases and development. Recent advances in single-cell multi-modal sequencing technologies make it possible to jointly profile chromatin accessibility and transcriptome at the same cell [4, 5, 6]. Through computational analyses of these multi-omics data, it becomes feasible to investigate the regulatory mechanisms governing cellular processes by establishing links between changes in chromatin accessibility and variations in gene expression at the single-cell resolution. However, challenges like the curse of dimensionality, the poor quality of single-cell data especially for the scATAC-seq datasets, and the scarcity of paired multi-omics data limit their practical potential.

Computational methods for single-cell multi-omics data have emerged, focusing on acquiring joint embeddings for latent space alignment, modality translation, and modality matching. Notable among these methodologies, WNN [7] applies a weighted nearest-neighbor strategy to integrate multimodal single-cell data to obtain a total embedding for each cell. Similarly, totalVI [8] concatenates the multi-modal data together and adopts a variational auto-encoder framework to jointly model them with a united representation. Nonetheless, both of these approaches generate unified representations

---

*Equal Contribution.

NeurIPS 2023 AI for Science Workshop.

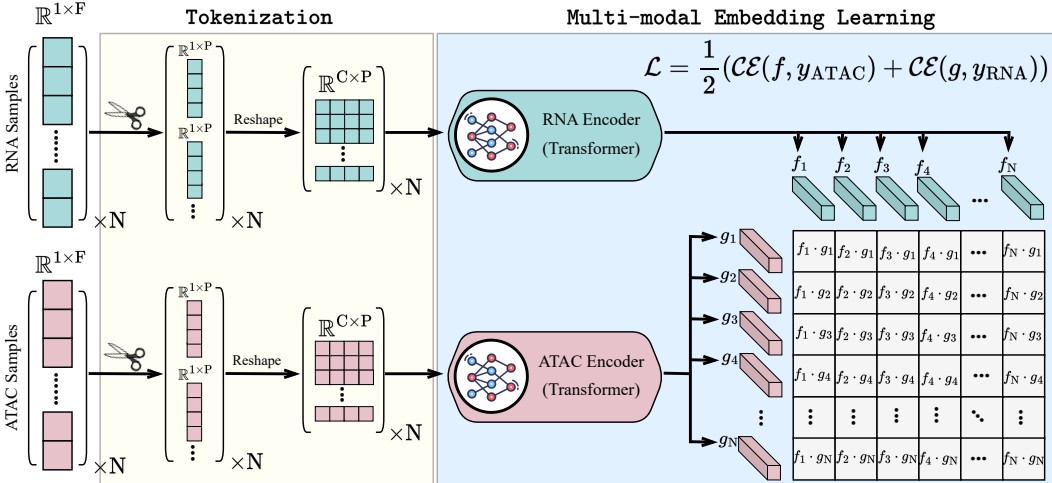

Figure 1: Overview of our framework. Single-cell profiles will be first processed by tokenization, and multi-modal embedding learning is conducted with generated input embeddings. $\mathcal{L}$ is the training objective as described in Equation 1.

instead of individual joint embeddings for each modality. This can potentially result in an uneven distribution of contributions from various modalities, often leading to the dominance of scRNA-seq data. Alternatively, LIGER [9] employs integrative non-negative matrix factorization (iNMF) to decouple the shared and dataset-specific factors. By doing so, it optimizes the joint embeddings of multimodal single-cell data within the shared space. In a parallel vein, BABEL [10] leverages a cross-auto-encoder model to predict single-cell ATAC from scRNA-seq data and vice versa. However, direct predictions of high-dimensional profiles typically prioritize highly variable features and inadvertently lead to reduced accuracy in predicting the majority of non-highly variable features. This effect is particularly pronounced for features that are crucial for rare cell populations. Beyond the realm of joint embeddings and modality translation, modality matching between multi-omics data is also a very important task, which enables querying the most similar data from the other modality. In the NeurIPS Competition'21[2], CLUE [11] method based on GLUE [12] achieved the best in modality matching task by linking different modalities using prior information and projecting multi-modal data into the same embedding space in a supervised way. A competitive method called Novel [3] proposed a contrastive framework and optimized bipartite graph for pairing match, which also achieved the top results in the competition. Recently developed MatchCLOT [13] combines contrastive learning and optimal transport match scATAC-seq and scRNA-seq also achieves good performance. However, these methods do not exhibit scalability or transferability when confronted with large-scale datasets of millions or even greater sizes.

In the machine learning community, Transformers [14] as a popular option has been frequently adopted by plenty of natural language processing (NLP) and computer vision (CV) applications with prevailing successes for large-scale pretraining [15, 16, 17, 18, 19, 20, 21, 22, 23, 24]. Especially, CLIP [25], a transformer-based multi-modal model, and its follow-up works have achieved impressive results on image and text integration tasks by learning a superior cross-modality matching. However, due to the high dimensionality of single-cell data, it is very difficult to tokenize the single-cell profile into an acceptable number of tokens, there are several successful trials applying transformer to single-cell RNA-seq datasets [26, 27, 28, 29]. the complexity and high dimensionality of scATAC-seq data, stemming from the large number of features typically ranging from hundreds of thousands to millions, present significant challenges in effectively harnessing Transformer models. To our knowledge, there have been no successful applications of Transformer models on scATAC-seq data.

In this study, we introduce a pioneering solution, *i.e.*, Contrastive Learning Integration Pre-training, named scCLIP for multi-modal single-cell data, aiming to address these challenges head-on. Our approach is the first feasible way to encode scATAC-seq data with Transformers to the best of our

---

[2]Multimodal Single-Cell Data Integration (https://openproblems.bio/events/2022-08_neurips/).

[3]Novel (https://github.com/openproblems-bio/neurips2021_multimodal_topmethods/tree/main/src/predict_modality/methods/novel)

knowledge, extending the same strategy to scRNA-seq data. By joint training two transformer-based encoders on paired multi-modal single-cell data and optimization using contrastive loss, `scCLIP` demonstrates its potential in accurately projecting multi-modal data into a unified embedding space and exhibits the capability to integrate multiple tissues and organisms at the large scale of atlas data. This enables the integration of different modalities within a joint embedding framework and facilitates the learning of transferable features across modalities. Overall, `scCLIP` represents a powerful tool for the integration of multi-modal single-cell sequencing data. Its ability to generate joint embeddings, promote transferability, and enable Atlas-level applications makes it a valuable asset in advancing our understanding of complex biological systems and their underlying regulatory mechanisms.

## 2 Methodologies

**Tokenization.** Tokenization as the necessary step to deal with high-dimensional inputs plays a crucial role in numerous successes achieved by transformer-based models [14, 30]. Inspired by the design in [30], we customize the tokenization for single-cell profiles. Specifically, a single-cell profile vector $x \in \mathbb{R}^F$ is split into C patches and each patch is a vector in $R^P$ where $F = C \times P$. To make each token have biological meanings, we sorted the features of ATAC-seq along the locations on the genomes, which means each token represents a segment of chromosome status. Note that zero padding will be performed to make each token have the same length. Then, these patches are fed into a transformer encoder, as indicated in Figure 1. In this way, patches from a single-cell profile are treated the same way words/tokens in NLP applications. Meantime, an extra "classification token" is learned as the global representation for each cell. Positional embeddings are introduced to capture the relative positional information, where 1D positional embeddings are considered in our case.

**Multi-modal Embedding Learning.** Motivated by the popular cross-modal learning algorithm, CLIP [25], we customize and propose our framework - scCLIP, as shown in Figure 1. Given a batch of N (ATAC, RNA) pairs, scCLIP is trained to predict which of the $N \times N$ possible (ATAC, RNA) pairings across a batch actually occurred. Vanilla transformers equipped with our tokenization are adopted to extract ATAC and RNA features. Both obtained feature embeddings are then projected to the latent space with an identical dimension. In this way, scCLIP learns a multi-modal embedding space by jointly optimizing an ATAC encoder and RNA encoder.

The objective is to maximize the cosine similarity of the ATAC and RNA embeddings from the N real pairs in the batch while minimizing the cosine similarity of the embeddings from the $N^2 - N$ incorrect pairings. Specifically, the loss function is described as follows:

$$\mathcal{L} = \frac{1}{2} \times (\mathcal{CE}(f, y_{\text{ATAC}}) + \mathcal{CE}(g, y_{\text{RNA}})), \tag{1}$$

where $\mathcal{CE}(\cdot, \cdot)$ is the cross-entropy function, $f$ and $g$ are the embeddings from RNA and ATAC profiles respectively, $y_{\text{ATAC}}$ and $y_{\text{RNA}}$ are their corresponding pairing labels.

**A Translational Matching.** An extra linear mapping function is learned to project scATAC embedding onto scRNA embedding space. In our case, the linear function is parameterized by a weight matrix W and a bias vector b to capture the rotation and translation shifts respectively.

## 3 Experiment Results

### 3.1 Implementation Details

**Evaluation Metrics.** Following the standard in [11], we choose `Matching Score` and `FOSCTTM` to evaluate our approaches, as depicted below.

▷ `Matching Score`. The average confidence in the correct pairing is determined by creating a cross-modality matching matrix $M_{i,j}$, which is constructed by calculating the Jaccard index of cross-modality nearest neighbors in the latent embedding space.

$$M_{i,j} = \frac{|(K_{ij} \cap K_{jj}) \cup (K_{ji} \cap K_{ii})|}{|(K_{ij} \cup K_{jj}) \cup (K_{ji} \cup K_{ii})|}, \tag{2}$$

where $K_{ij}$ represents the nearest neighbor of the sample $i$ in the modality of a sample $j$.

$$\texttt{Matching Score} = \frac{1}{N} \sum_i \sum_j \tilde{M}_{i,j} * \delta_{i,j}. \tag{3}$$

The `Matching Score` (MS) is then calculated according to the formula 3, where $\tilde{M}$ is obtained by normalizing the matching matrix $M$ on a per-row basis, $\delta_{i,j}$ equals 1 if profile $i$ and $j$ were measured in the same cell and 0 otherwise. $N$ represents the total number of observations.

▷ `FOSCTTM`. Fraction of Samples Closer than True Match (`FOSCTTM`) [31]. $N$ is the number of cells. $n_1^{(i)}$ and $n_2^{(i)}$ denote the number of cells that are closer to the cell $i$, compared to their true matches in the opposite dataset. The distance function $d(\cdot, \cdot)$ considered here is the Euclidean distance. To be specific:

$$\texttt{FOSCTTM} = \frac{1}{2N} \left( \sum_{i=1}^{N} \frac{n_1^{(i)}}{N} + \sum_{i=1}^{N} \frac{n_2^{(i)}}{N} \right), \tag{4}$$

where $n_1^{(i)} = |\{j|d(x_j, y_i) < d(x_i, y_i)\}|$, $n_2^{(i)} = |\{j|d(x_i, y_j) < d(x_i, y_i)\}|$. Note that lower `FOSCTTM` values imply higher accuracy.

▷ `Adjusted Rand Index (ARI)`. The Rand Index (RI) computes the similarity score between two clustering assignments by considering matched and unmatched assignment pairs independently of the number of clusters. The Adjusted Rand Index (ARI) score is calculated by "adjust for chance" with RI by:

$$\texttt{ARI} = \frac{\text{RI} - \text{Expected\_RI}}{\max(\text{RI} - \text{Expected\_RI})} \tag{5}$$

▷ `Normalized mutual information (NMI)`.

$$\texttt{NMI} = \frac{\mathcal{I}(P, T)}{\sqrt{\mathcal{H}(P)\mathcal{H}(T)}} \tag{6}$$

where $P$ and $T$ are empirical categorical distributions for the predicted and real clustering, $\mathcal{I}$ is the mutual entropy, and $\mathcal{H}$ is the Shannon entropy.

▷ `Silhouette Score`. The silhouette score was computed by combining the average intra-cluster distance (a) and the average nearest-cluster (b) for each cell.

$$\texttt{Silhouette Score} = \frac{b - a}{\max(a - b)} \tag{7}$$

▷ `Batch entropy mixing score`. Batch entropy mixing score was used to access the regional mixing of cells from different batches or between reference and query datasets, with a high score suggesting that cells from different batches are well mixed together. The batch entropy mixing score was computed as follows:

(1) Calculate the proportion $P_i$ of cell numbers in each batch to the total cell numbers.
(2) Randomly choose 30 cells from all batches.
(3) Calculate the 30 nearest neighbors for each randomly chosen cell.
(4) The regional mixing entropies for each cell were defined as:

$$p_i' = \frac{\frac{p_i}{P_i}}{\sum_{i=0}^{n} \frac{p_i}{P_i}}, \quad \mathcal{E} = \sum_{i=0}^{n} p_i' \log(p_i') \tag{8}$$

where $p_i$ is the proportion of cells from batch $i$ in a given region, such that $\sum_{i=0}^{n} p_i = 1$, $p_i'$ is a correction item to eliminate the deviation caused by the different cell numbers in different batches. The total mixing entropy was then calculated as the sum of the regional mixing entropies.

(5) Repeated (2)-(4) for 10 iterations with different randomly chosen cells and calculated the average, $\mathcal{E}$, as the final batch entropy mixing score.

Note that to mitigate the effect of misalignment of batch-specific cell types, we calculated the batch entropy mixing score only based on cells from cell types that are common in different batches.

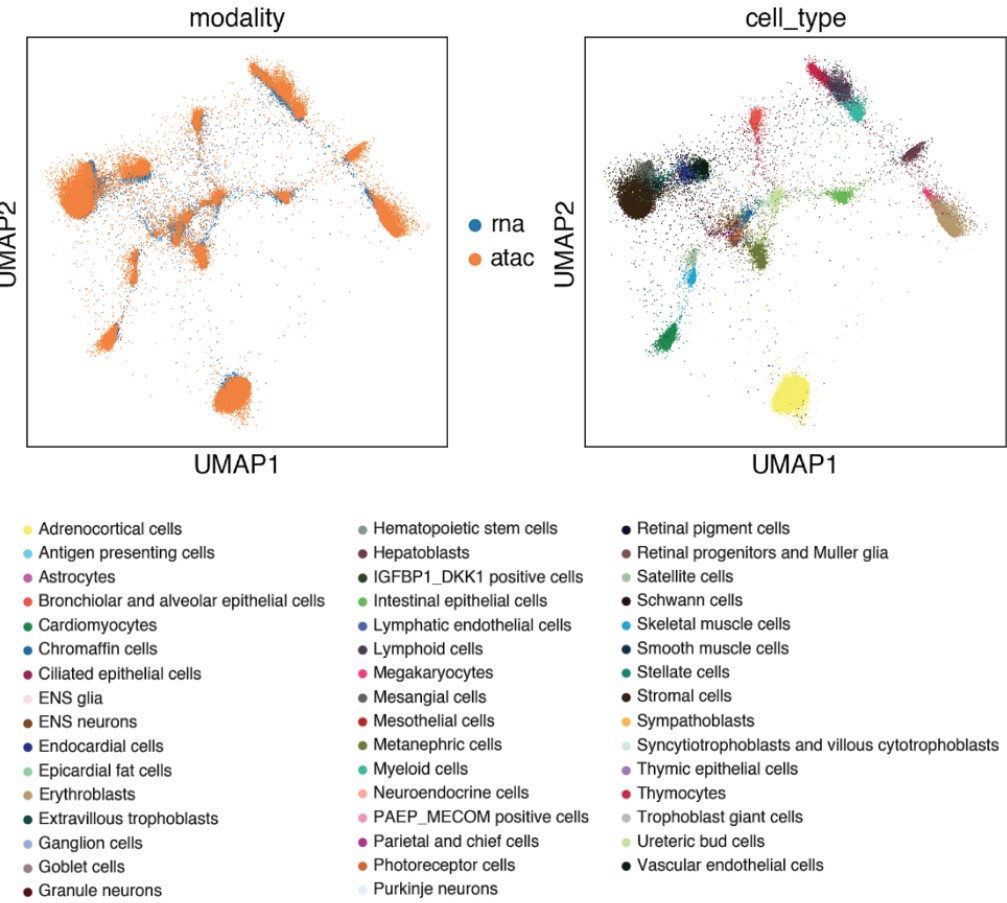

Figure 2: Visualization of RNA and ATAC joint embedding on UMAP. Each point represents a cell, Each point represents a cell, and only the validation data is included in the visualization. The left panel is colored based on modality, while the right panel is colored based on cell type. The legend for cell types is located at the bottom. Please zoom in for better readability.

**Datasets**    *Fetal Atlas.* Since there is no available Atlas-level paired single-cell multi-omic data, we instead collected single-cell atlas of fetal gene expression [32] and chromatin accessibility [33]. These two datasets were generated from the same lab with the same organs and consistent annotations, we manually curated a pseudo-paired dataset by randomly paired cells within the same cell annotations. Within this dataset of $377,134$ cells, the scATAC-seq data consists of $1,154,464$ peaks, while the scRNA-seq data comprises $36,601$ genes. *Brain.* We examine our method on a paired AD (Alzheimer's disease) of human brain tissue dataset [34] and apply it to another paired dataset downloaded from the 10x Genomics website[4]. In order to enhance the generalizability of our model, we retain all features without selecting highly variable genes or filtering any peaks, following the common approach employed in single-cell methods.

**Transferable Evaluation**    To investigate the transferability, we randomly split each dataset into training and validation with a ratio of $9:1$, and train the scCLIP model on the paired training datasets. Later on, the converged models are directly evaluated on the validation split and other unseen datasets that might be paired or unpaired. During the transfer evaluation, ATAC and RNA profiles are projected separately using the trained encoders to the same embedding space, and their alignment consistency is measured to indicate the performance.

---

[4]`https://cf.10xgenomics.com/samples/cell-arc/2.0.0/human_brain_3k/human_brain_3k_filtered_feature_bc_matrix.h5`

Table 1: Metrics for cell type separation and modality alignment in population level. All the metrics are calculated with UMAP embeddings. Silhouette score is used to evaluate cell separation with cell type annotations as labels. Batch entropy mixing score is used to evaluate modality alignments with modality information as labels. ARI and NMI are used to evaluate clustering accuracy by comparing the Leiden cluster results with cell type annotations.

| Method | Silhouett score | Batch entropy mixing score | ARI | NMI |
|---|---|---|---|---|
| scCLIP | 0.486 | 0.578 | 0.780 | 0.813 |
| CLUE | 0.201 | 0.595 | 0.713 | 0.263 |
| MatchCLOT | 0.414 | 0.439 | 0.826 | 0.849 |

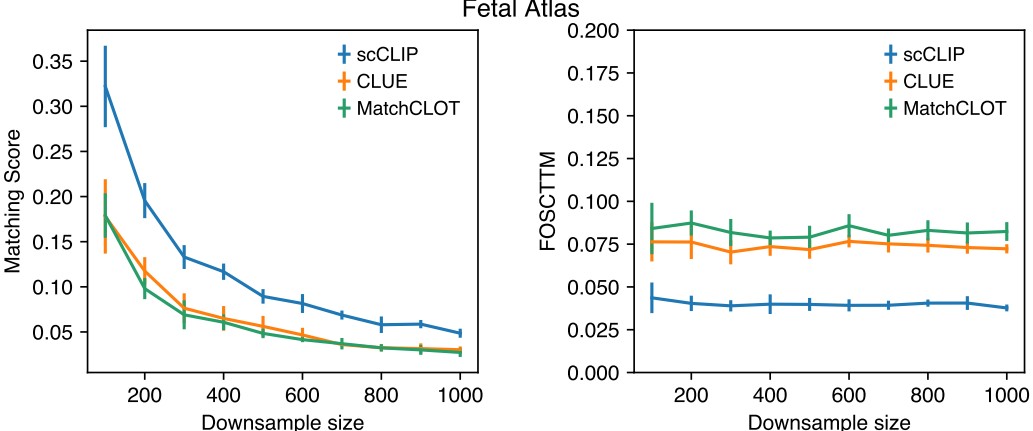

Figure 3: Comparison of `Matching Score` and `FOSCTTM` metrics. The metrics to evaluate the performance of scCLIP and CLUE are calculated by randomly downsampling to various sample sizes ranging from 100 to 1000 with an interval of 100. The Matching Score, where higher values indicate better performance, and FOSCTTM, where lower values indicate better performance. It is important to note that as the sample size increases, the evaluated scores may decrease due to the similarity of single-cell data within the same cell type and the data scarcity.

## 3.2 Experiment Results

We conducted initial validation of our model using the *Fetal Atlas* dataset to showcase its exceptional scalability for Atlas-level datasets, as well as its ability to accurately cluster cell populations and align different modalities. The evaluation results were generated using solely the validation data, which was unseen by the model during training. In Figure 2, The UMAP visualization not only showcases the consistent alignment of RNA and ATAC embeddings achieved by `scCLIP`, but also reveals distinct separation of cell populations, with each color representing a different cell type.

When comparing scCLIP to other competing methods such as CLUE and MatchCLOT, based on the UMAP embeddings, scCLIP attains the highest silhouette score. This signifies that scCLIP effectively preserves biological variances by accurately separating distinct cell populations, as indicated in Table 1. Additionally, the high ARI and NMI scores further corroborate scCLIP's proficiency in this regard. In terms of population alignment, scCLIP also achieves comparable results to CLUE in terms of batch entropy mixing scores between the two modalities, even though scCLIP focuses solely on optimizing individual pairs of cells without utilizing any population-based information.

Then, we employed the `Matching Score` and `FOSCTTM` metrics to quantitatively assess the correspondence between the two modalities, comparing our approach with the state-of-the-art method CLUE. Figure 3 clearly illustrates that scCLIP consistently outperforms CLUE and MatchCLOT across varying sample sizes, exhibiting a substantial improvement of nearly 80% improvement in Matching Score (higher values indicate better performance) and improvements from approximately 0.08 to a more precise 0.04 in `FOSCTTM` (lower values indicate better performance) comparing to CLUE and MatchCLOT.

Subsequently, we demonstrated the generalizability of our model by showcasing its ability to learn transferable features across datasets, a capability lacking in other competing methods. Specifically,

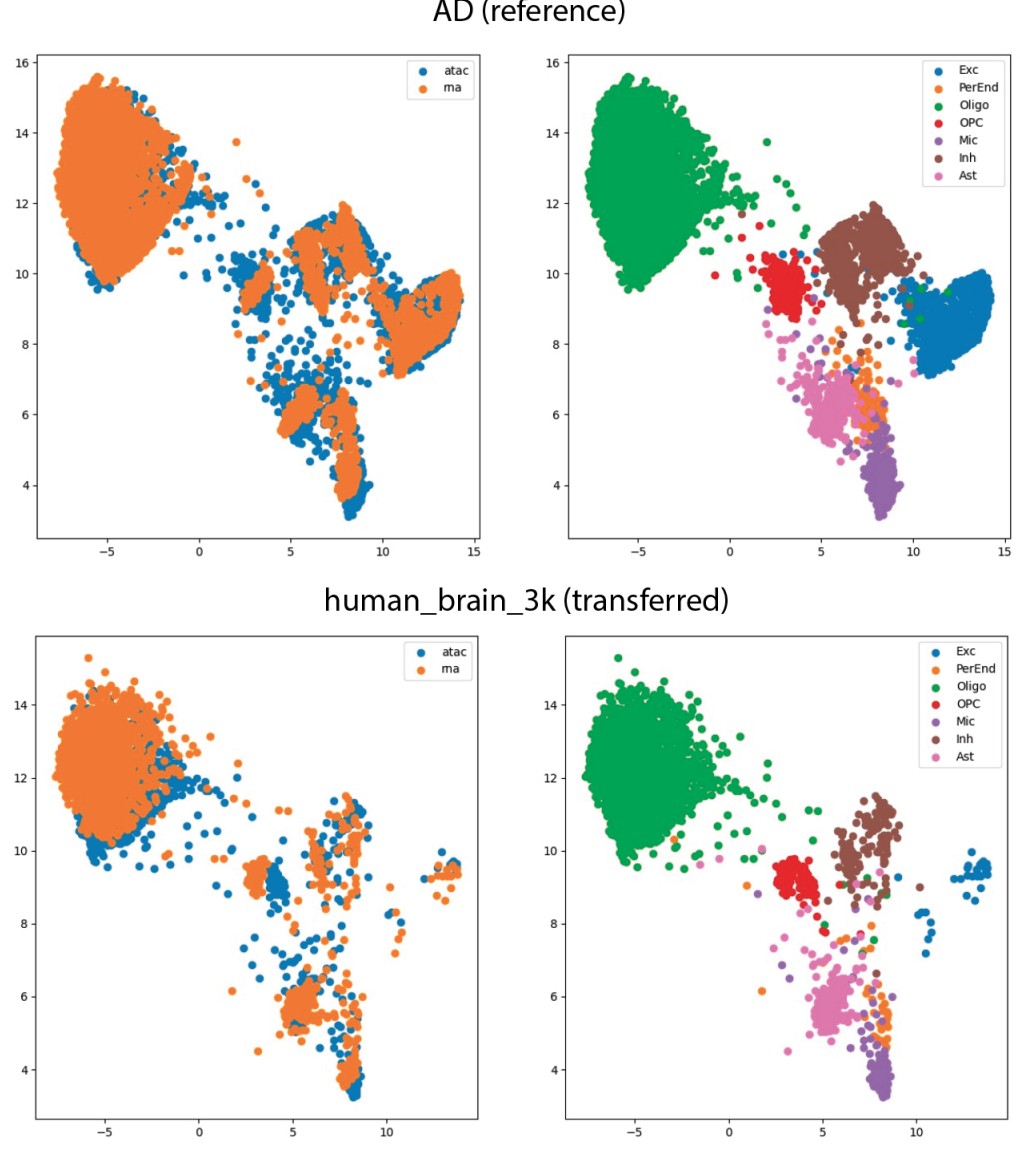

Figure 4: Visualization of the joint embedding of RNA and ATAC on UMAP. (Top) The RNA embeddings and the ATAC embeddings from AD (reference) align well. (Bottom) The embeddings of the *human_brain_3k* (transferred) dataset are projected onto the same space as AD. The left panel is colored based on modality, while the right panel is colored based on cell type. The cells of the same cell types are represented with the same color.

we trained the model on the *AD* dataset and directly applied it to the *human_brain_3k* dataset without the need for additional training. Figure 4 visually depicts the results, showing that the embeddings of scATAC-seq and scRNA-seq data from the AD dataset (with validation data displayed) are well integrated, exhibiting a good alignment between the two modalities and clear separation of cell types. Notably, the transferred *human_brain_3k* data not only demonstrates strong alignment within its own modalities and cell types but also aligns with the cell embeddings of the *AD* dataset. This alignment is evident in the UMAP plot, where the *AD* embeddings and *human_brain_3k* embeddings occupy the same positions, indicating the presence of shared or similar cell populations. These findings indicate that our model has the ability to learn transferable features related to cell type across datasets in both scATAC-seq and scRNA-seq.

# 4 Discussion

Our experiments demonstrate that employing a patch-based tokenization approach with transformers on single-cell profiles can yield effective representations. To the best of our knowledge, this marks the pioneering application of the transformer model to scATAC-seq data, thus expanding the possibilities for leveraging transformer-based approaches in single-cell analysis. Through joint optimization, we discern transferrable features across modalities, establishing a connection to align scATAC-seq and scRNA-seq data. This connection enables us to gain a more comprehensive understanding of the intricate interplay between chromatin accessibility and gene expression. Additionally, by working with Atlas-scale datasets, we are able to delve deeper into the complexities of this interplay.

Our work has certain limitations. We attempted to directly align different modalities to the same embedding space by adjusting without the need for an additional translation layer. However, this approach does not generalize well to all datasets, particularly smaller ones, as the model tends to overfit in such cases. In future work, we plan to explore incorporating pre-trained models trained on large-scale multiomics data from HuBMAP Consortium `https://hubmapconsortium.org/` as well as unimodal single-cell data and fine-tuning them using either paired datasets or unimodal data with iteratively refined labels. Simultaneously, we will investigate a more biologically meaningful approach to patch genes and peaks, enabling a direct link between features from different modalities and making the transferable features more interpretable.

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
