# OpenReview forum: "scCLIP: Multi-modal Single-cell Contrastive Learning Integration Pre-training"
_NeurIPS.cc/2023/Workshop/AI4Science — NeurIPS2023-AI4Science Poster_

### Official Review · Reviewer_3RRR · 2023-10-20
**An interesting paper about utilizing CLIP to model single-cell data**

**Rating:** 7
**Confidence:** 4

**Review:**

In this manuscript, the authors offer a CLIP based model for the paired multi-omics data integration task. They prove the supuority of their method by comparing the batch related metrics and biological conservation related metrics. Here I have some questions to ask:

1. In the dataset selection step, it is hard to conclude that we do not have atlas level multiomics data. Please check the HUBMAP project page for atlas-level single-cell multiomics data: https://commonfund.nih.gov/HuBMAP. I think it will more interesting if the authors can discuss the data ablation proces for this task.

2. There are also other methods like CVQVAE [1] and scMoGNN [2]. What is the performance difference betwen these two methods and scCLIP?

I am looking forward to see your improvement. Nice work!


[1] https://proceedings.mlr.press/v200/liu22a.html
[2] https://dl.acm.org/doi/pdf/10.1145/3534678.3539213

---

### Official Review · Reviewer_rsJj · 2023-10-24
**I believe the paper is satisfactory, but it requires improvements in certain details of the writing to enhance clarity**

**Rating:** 6
**Confidence:** 5

**Review:**

I believe the paper is satisfactory, but it requires improvements in certain details of the writing to enhance clarity. Additionally, the results are promising.

1. **Prior Work:**
    - It's important to highlight that similar methods based on the CLIP approach were utilized by the Novel team during the NeurIPS 2021 single-cell multi-omics competition. This raises questions about the novelty and differentiation of the current approach. The authors are encouraged to make explicit comparisons and highlight unique contributions.

2. **Scalability (Line 52):**
    - The paper claims certain methods cannot be scaled to millions of cells. This is perplexing, considering most previous techniques are deep learning-based and, when fixed with the right batch size, have showcased scalability. An explanation or justification for this claim would add clarity.

3. **Data Patching (Line 82):**
    - The methodology of splitting the patches is not clearly described. Given that gene order in the single-cell profile can be somewhat arbitrary, and the same goes for ATAC-seq, how are these challenges addressed? A detailed methodological explanation is crucial here.

4. **Classification Token:**
    - The mention of an "extra classification token" is a bit ambiguous. A clear explanation of its role and utility would help in understanding the model architecture and its benefits.

5. **Translation Matching Linear Mapping:**
    - The motivation behind introducing translation matching linear mapping is not evident. Why not use just contrastive learning? Is there a specific domain gap that this mapping addresses? Clarifying this would add depth to the paper.

6. **Selection Methodology (Line 135):**
    - The reason behind selecting only 30 cells from all batches is unclear. Justifying this selection criterion is crucial to understand the representativeness and robustness of the approach.

7. **Data Splitting (Line 156):**
    - The choice to split data randomly does not reflect real-world scenarios. It would be more rigorous and realistic to split by batches, ensuring that the evaluation is robust against batch effects and other possible sources of variance.

---

### Meta-Review · Area_Chair_Emnd · 2023-10-27

**Recommendation:** Accept (Oral)
**Confidence:** 3

**Metareview:**

The paper presents interesting work with potential for improvement in specific areas. While the results are promising, some questions should be addressed. The authors are encouraged to provide detailed comparisons and emphasize the unique contributions of their work. A more comprehensive comparison of the proposed method, highlighting its distinctions and performance compared to other methods, and more clarity on the limitations or drawbacks of different approaches mentioned in the introduction would significantly improve the paper. Furthermore, as noted by the reviewers, several technical aspects of the method require further clarification in the paper. The authors should also provide additional evidence to support their claims regarding the applicability of their method to real-world data.